# High Performance Marine and Terrestrial Bioadhesives and the Biomedical Applications They Have Inspired

**DOI:** 10.3390/molecules27248982

**Published:** 2022-12-16

**Authors:** James Melrose

**Affiliations:** 1Raymond Purves Bone and Joint Research Laboratory, Kolling Institute, Faculty of Medicine and Health, University of Sydney at Royal North Shore Hospital, Northern Sydney Local Health District, St. Leonards, NSW 2065, Australia; james.melrose@sydney.edu.au; 2Graduate School of Biomedical Engineering, University of New South Wales, Sydney, NSW 2052, Australia; 3Sydney Medical School, Northern Campus, The University of Sydney, St. Leonards, NSW 2065, Australia

**Keywords:** mussel, barnacles, caddis-fly fly, gecko, slug, marine bioadhesive, spider silk, fibroin, sericin, seroin, L-DOPA, catechol

## Abstract

This study has reviewed the naturally occurring bioadhesives produced in marine and freshwater aqueous environments and in the mucinous exudates of some terrestrial animals which have remarkable properties providing adhesion under difficult environmental conditions. These bioadhesives have inspired the development of medical bioadhesives with impressive properties that provide an effective alternative to suturing surgical wounds improving closure and healing of wounds in technically demanding tissues such as the heart, lung and soft tissues like the brain and intestinal mucosa. The Gecko has developed a dry-adhesive system of exceptional performance and has inspired the development of new generation re-usable tapes applicable to many medical procedures. The silk of spider webs has been equally inspiring to structural engineers and materials scientists and has revealed innovative properties which have led to new generation technologies in photonics, phononics and micro-electronics in the development of wearable biosensors. Man made products designed to emulate the performance of these natural bioadhesive molecules are improving wound closure and healing of problematic lesions such as diabetic foot ulcers which are notoriously painful and have also found application in many other areas in biomedicine. Armed with information on the mechanistic properties of these impressive biomolecules major advances are expected in biomedicine, micro-electronics, photonics, materials science, artificial intelligence and robotics technology.

## 1. Introduction

This study reviews a number of naturally occurring adhesive molecules produced in marine and terrestrial environments and some of the biomedical applications which have been inspired by these. Nature’s bioadhesives perfected through positive natural evolutionary selection pressure over the millennia has produced some remarkable, innovative, sticky solutions, to some extremely difficult situations far surpassing the performance of man-made adhesives. Bioadhesives that securely attach filter feeding molluscs such as mussels and barnacles to support structures underwater are two good examples of these involving a collection of adhesive molecules which act in a coordinated manner using a number of chemistries to firmly and in some cases permanently cement these molluscs in place under wet and extremely adverse binding conditions (Figure 1). This binding process occurs rapidly in a hostile marine environment under wet conditions. A better understanding of these adhesive processes has received considerable attention since such strong and robust wet adhesives are of potential application in demanding areas of medicine to re-attach heart, liver, neural and soft mucosal tissues following surgical interventions and in orthopaedic procedures which require high performance adhesives which can withstand high weight bearing and tensional stressful environments.The use of an adhesive in these procedures offers several technical advantages over suturing and improved outcomes for the patient.

The marine Sandcastle worm (*Phragmatopoma californica*) (Figure 1d–f) and freshwater Caddis-fly (order *Trichoptera*) larvae (Figure 2) precisely assemble protective encasing tube-like structures using an adhesive to cement particles of sand, crushed crustacean shell, vegetation or small stones into a protective encasing structure (Figure 2d–f). This adhesion process has been examined in-vitro using small glass beads illustrating the high precision and structural features of this tube assembly process (Figure 2a–c). Furthermore, such Sandcastle worm formations also produce higher order tube-networks in marine environments which stabilize sand reef formations preserving the integrity of the shoreline from erosion. The Caddis-fly tube attaches a similar tube-like structure to the underside of submerged large stones where the Caddis-fly larvae are protected from predators and they undergo metamorphosis to the nymph stage prior to their emergence from these tube-like structures as mature insects (Figure 2g,h).

Some surprising terrestrial organisms (snail, slug) utilize innovative adhesives to provide firm but selective bimodal attachment to surfaces to not only facilitate sliding movement over surfaces, but these exudates under certain circumstances can also become highly adhesive providing firm anchorage to surfaces that provides protection from their removal by predators. Green tree frogs inadvertently encountering such exudate deposits from red triangle slugs on tree branches remain immobilised for several days (Figure 3a,b). Some of these exudates have even been shown to have anti-bacterial properties. A particularly unusual aspect of the dorsal skin exudate of the crucifix toad *Notaden benetti* is its semi-permanent adhesive properties which ensures prolonged sexual union occurs over many hours ensuring mating has a successful outcome (Figure 3c,d). In some reptiles (gecko, salamander) and amphibians (frog) the adhesive properties of mucous secretions on their tongues are differentially regulated to improve capture of food species by the tongue but become non-adhesive when the captured item is recovered into the mouth (Figure 3e,f). The mucilaginous exudate of slugs has impressive adhesive properties in some contexts and has been developed into a medical adhesive (Figure 3g–j). This is discussed further later in this review.

Research on bio-adhesives has mainly focussed on marine sources, bioadhesives produced by terrestrial animals are relatively rare [3,4,5]. Several reptilian species have developed a sticky mucus on highly extensible projectile elastic tongues which have wet-adhesive properties which aid in the capture of food species [6,7]. Moreover the properties of the mucous salivary secretions which coat the tongue have bimodal properties and can be varied from a low adhesive lubricative form to one which has adhesive properties which firmly attach prey species to ensure efficient capture [8,9,10,11,12,13]. Upon capture, the saliva reverts to a low adhesive form which allows release of the food item in the mouth. Frog tongue saliva is a two-phase viscoelastic fluid, the two phases are dependent on how quickly the saliva is sheared, when resting between parallel plates. At low shear rates, the saliva is very thick and viscous and strongly adhesive but at high shear rates, the frog saliva behaves more like a non-adhesive ideal Newtonian fluid with thixotropic properties [6,8,10]. Several adhesive reptilian skin exudates have also been described. One particularly interesting one is the exudate of the dorsal skin of the crucifix toad *Notaden benetti*. This has strong adhesive properties and ensures that the male remains anchored to the female during sexual union for prolonged periods ensuring the success of the sexual activity [3,4,14,15]. The bonding strength of terrestrial glues (up to 1.7 MPa for the Australian frog genus Notaden) are amongst the highest in the animal kingdom and of comparable adhesive strength to cyanoacrylates but have superior material properties under wet conditions and are non-toxic and do not elicit an immune reaction [16]. When provoked by predators *Notaden bennetti* secretes a sticky adhesive from its dorsal skin deterring their consumption as a food item, this adhesive also prolongs sexual union during mating [3]. This secreted mucous exudate rapidly transforms into an elastic hydrogel with adherent properties for a wide range of materials, including glass, plastic, metal and even Teflon [16]. These secretions are mainly proeinaceous and contain little carbohydrate. The high bonding strength of the Notaden glue has generated interest in commercial applications including orthopaedic meniscal and tendon reconstructions [17,18] and in the industrial [5,19] and medical sectors.

The dusky and Australian red triangle tree slug also exhibits a multi-modal mucinous exudate which they utilize as a lubricant to slide over surfaces and as an adhesive sticky secretion which has impressive adhesive properties and can immobilize potential predators in-situ when the slugs are threatened. The impressive adhesive properties of this slug slime became apparent when slime deposited by the Australian red triangle tree slug was found to immobilize red eyed green tree frogs on tree branches for several days [20]. The slime exuded by the dusky slug has inspired the development of an innovative surgical bioadhesive which has been applied in abdominal and heart surgery and has impressive performance credentials [21]. This glue may obviate the use of suturing in heart surgery which can be time consuming and is a potential source of infection with the suture holes leaking blood and or serum products for extended periods after surgery and has been used to glue patches over heart defects. The beating heart is a particularly demanding target tissue and the successful application of this glue in heart surgery amply illustrates its impressive credentials.

Some species of lizards and frogs have evolved fascinating dry adhesive pad-like structures which utilise microfluidic processes to manipulate frictional forces generated by microscopic tiered nano-fibre hair-like projections (setae) in their foot pads to provide adhesion on smooth surfaces [22]. The extraordinary performance of these dry adhesives have been a source of fascination and inspiration to scientists and engineers for over a century and have led to the development of novel commercial adhesive products in the form of new g generation high performance adhesive tapes and textile materials with enhanced gripping properties (Figure 4). The strapless and backless push-up bra is a notable application of this technology, however gloves used in baseball and by soccer goal-keepers have also been developed with superior gripping properties. The gecko adhesion system and its strong, reversible, self-cleaning properties are impressive credentials which scientists and engineers have attempted to emulate in commercial products [23]. Some frogs have also evolved similar adaptations to the ghecko adhesive system in their foot parts to optimize their gripping properties in their arboreal environments [24]. Scientific studies on the extraordinary adhesive properties displayed by the foot parts of geckos was first reported almost three decades ago [25]. Ghecko’s use a complex hierarchical adhesive system which utilises the maximized surface area of Ghecko feet and their nanoscale fibres (setae) to produce adhesion through van der Waals forces with the attachment and detachment phases of this process controlled by the loading angle of the nano-fibres in their foot parts [26] (Figure 4). Considerable research has been devoted to creating gecko inspired adhesives with desirable dry adhesive properties [27,28,29].

A re-usable superadhesive tape has also been developed inspired by the adhesive properties of the ghecko and tree-climbing frogs. Conventional adhesive tapes cannot be re-used since micro-cracks are introduced into the adhesive interface when these tapes are peeled from adherent surfaces. These cracks weaken the adhesion of such tapes allowing their removal but renders them non re-usable. The adhesive on the toe pads of tree frogs and ghecko’s contain microscopic channels that prevent cracking when they are peeled from a surface. Furthermore addition of fluid to these microchannels increases their adhesive strength by as much as 30-fold. Because adhesion occurs when the molecules on two surfaces are close enough to allow intermolecular attraction to occur, this increases the “stickiness” of the adhesive through capillary action and since the adhesive is elastic it is also re-usable. Superadhesive tapes have been developed for use in surgical applications in orthopaedics [31,32,33,34,35] and in soft tissue reconstruction procedures [36,37,38,39,40,41,42]. Novel surgical tapes containing BMP-2 binding peptides have been used to capture endogenous growth factors to accelerate healing processes in anterior cruciate ligament reconstructions [43].

The perfection in architectural form displayed by spider silk webs using minimal amounts of structural material and the extraordinary mechanical properties of constituent silk proteins used to assemble these structures has inspired scientists, structural engineers and architects for over a thousand years who have attempted to develop polymers which emulate the superior performance of these spider silk proteins. The biomechanical performance of spider webs to extreme stressful environments has also provided important structure-function information to engineers [44]. Ornate spider web structures have provided insights into appropriate measures that are required in the design of man-made engineered super-structures to ensure their stability and load bearing properties [45,46,47]. The spider web adhesive proteins are discussed in more detail later in this review.

The structure of spider webs inspired the development of large radial cable network supported ultra light structures in the distinctive domed stadium complexes of The Munich Summer Olympics (1972) and in design features applied in the construction of major sporting stadiums in Frankfurt, Warsaw, Bucharest, Beijing Olympics (2008), Bao’an stadium, Shenzen (2011). London Olympic Stadium (2012) featured the longest cantilevered roof in the world. Innovative suspended radial cable supported roof structures were also used in Krasnodar stadium (2016), and have been used in the construction of The Qatar 2022 World Cup Soccer Stadium [48,49].

An important component which is integral to the assembly and stabilization of spider webs are the spider pyriform bioadhesives which anchor the web components to one another and the periphery of the web to peripheral stabilizing structures. Spiders also produce small deposits of adhesive on specific inner regions of the web sub-structure which facilitate the capture of prey species which alight on these internal regions of the spider web.

## 2. Mussel Foot Proteins and the Adhesive Process

Six mussel proteins (Mfp 1–6) are found in the adhesive plaques of *Mytilus edulis*, *Mytilus galloprovincialis*, *M. californianus* (Figure 5a). These are basic proteins containing various amounts of the unique catecholic amino acid, L-DOPA (l-3,4-dihydroxyphenylalanine), L-DOPA is biosynthesized from the amino acid, L-tyrosine by the enzyme tyrosine hydroxylase (Figure 5b). Mfp1–6 contain variable amounts of L-DOPA varying from 2–30% of their amino acid composition. The catechol side chain of L-DOPA offer robust and durable adhesion to various substrate surfaces and contribute to the curing of the adhesive plaque formations. Each of the Mfp’s have unique properties that collectively contribute to the byssus attachment. Mfp-1 is a 108 kDa protein in *M. edulis* which forms a protective cuticle like covering to the byssus threads. Mfp-2 is a 42–47 kDa protein in *M. edulis* containing highly repetitive motifs and is the most abundant byssus protein. Mfp-2 has a relatively high content of cysteine residues which are involved in stabilizing disulphide bond formations but has a low L-DOPA content. Byssal collagens (PreCols) distributed in the byssal thread have high L-DOPA and His contents. Mfp-3, 5, and 6 are predominantly found at the plaque–substrate interface, contributing to strong, wet adhesion. Mfp-3, the smallest adhesive plaque protein (5–7 kDa in *M. edulis*) is devoid of repeat sequences however 30–35 variants have been reported with variable L-DOPA incorporation levels. Mpf-4 interacts with the His-rich domain of precols through metal ion coupling. Mfp-5 is a 8.9 kDa protein containing the highest amount of L-DOPA (30 mol %) of any of the plaque proteins and also contains variable levels of phosphoserine residues. The high L-DOPA content of Mfp-5 and its phosphoserine content indicate that Mfp-5 has important binding roles at the plaque-substrate interface. Mfp-5 is the most adhesive protein of the mfps however, its adhesion, at least to mica, is highly reliant on low pH and the absence of oxidants. Another byssal protein, Mfp-6 contains cysteine residues which rescues adhesion of Mfp-3/5 by reducing L-DOPAquinone back to L-DOPA maximizing its adhesive properties [50]. This facilitates *Mytilus edulis* Mfp-5 byssal adhesive plaque adhesion to sub-strata, its high 3,4 L-DOPA content (~30 mol %) and localization near the plaque-substrate interface indicate a key role in adhesion. Mfp-5 also exhibits strong protein-protein interactions both with itself and with Mfp-3 improving plaque cohesion [51]. Mfp-5 from *M.edulis* is a small 9–10 kDa protein and besides its high L-DOPA content also contains an appreciable glycine (15 mol %) and lysine (17 mol %) content [52,53]. Basic residues are counteracted by glutamic acid and phosphoserine modifications producing a very hydrophilic protein. L-DOPA-based mussel adhesion is regulated through catecholic (reducing) and quinonic (oxidizing) chemistries contributing to the maturation of the byssal attachment. L-DOPA residues may undergo crosslinking either through auto-oxidation of L-DOPA to L-DOPAquinone or through the action of catechol oxidase [54]. The high L-DOPA content of Mfp-3 (20 mol %) and Mfp-5 (30 mol %) ensures surface protein adhesion (Figure 5c). Mfp-3 and 5 act as primers for the adhesive surface preparing it for the attachment of other mfps in a hierarchical temporal cascade of interactions. Besides the conversion of tyrosine residues to L-DOPA in Mfp-3/5, a significant number of serine residues in Mfp-5 are also phosphorylated [52]. This may be an adhesive adaptation to enhance binding to calciferous surfaces, phosphoserine cement proteins are also a prominent feature of barnacle, Sandcastle worm and oyster adhesion. Additional interactive processes with the quinolone functionalities of L-DOPA may also contribute to adhesion through surface chemistries involving (i) electrostatic interactions, (ii) hydrogen bonding, (iii) hydrophobic interactions, (iv) cation–π, and (v) π –π redox electron reorganization between benzene rings, and (vi) metal-coordination interactions [55] (Figure 5d). Catechol chelates metal ions to form strong, reversible complexes with Cu, Zn, Mn, Fe, V, Ti, of variable mono-, bis-, and tris-catecholate–metal ion complexes, depending on the valency of the metal ion. In addition to forming strong complexes with metal ions, catechol also forms strong, reversible interfacial bonds with metal oxide surfaces. The oxidation/reduction properties of the catechol functional motif in L-DOPA facilitates a reactive environment which can be precisely controlled to mediate adhesion and cross-linking in the mussel byssus foot-structure with the variable L-DOPA content of mfps providing a flexible system for the promotion of such adhesive processes.

Distinct L-DOPA-based cross-linking pathways are possible during byssus adhesive processes via oxidative covalent cross-linking or by formation of metal coordination complexes produced under reducing conditions. These processes are undertaken in spatio-temporal compartments of the byssus in which the proteins are stored prior to their secretion [56]. Metal ions (iron and vanadium taken up from seawater) are also released from specialized compartments and are believed to have roles in the maturation process and hardening of byssus glue components [57]. The mussel is one of only a few organisms that accumulate vanadium and use it in bioadhesive processes.

While 11 Mytilus byssal proteins are known with roles in plaque adhesion, a more recent in-depth proteomic study identified a further 33 byssal proteins [58], 17 of these were specific to the proximal thread and 5 to the distal thread. These displayed collagen-like, C1q domain-containing, protease inhibitor-like, tyrosinase-like, or shared homologies with SOD extending the repertoire of Mytilus byssal proteins, their specific roles in mussel adhesion will become apparent upon completion of structure function studies [58]. The byssal collagens (preCols) act as spring-like mechanical shock absorbers, imparting strength and stiffness as well as elastic properties. The byssal collagens do not have triple helical and cross-linking structures like those which stabilise mammalian fibrillar collagens but have Gly-Gly-X repeats and elastin-like regions which are important for byssal function, they also lack the quarter-stagger assembly of amino acids typical of fibrillar interstitial collagens. Metal-binding by multiple histidine residues may mediate inter- and intramolecular stabilization of preCols in the byssus [59].

Secure attachment of mussels to rocks is facilitated by byssal threads which have viscoelastic and energy dissipative properties and act as shock absorbers during cyclic loading [60,61]. Byssal threads combine high strength and toughness and are highly extensible (up to 200% extensible). These properties are attributable to multi-domain byssal collagenous proteins which are elastomeric block copolymers containing silk-like domains [62,63,64]. In the mussel byssus, reversible histidine-metal coordination is a key feature, that mediates higher-order self-assembly and self-healing of the acellular byssus thread [65]. This represents a versatile cross-linking mechanism which contributes to the mechanical properties of the byssus thread and is of potential application in the fine-tuning of the viscoelastic properties of biomimetic hydrogels [66]. The byssal collagens (preColls) of marine mussels are extracorporeal collagens that act as a shock absorber, these molecules are strong and stiff at one end and elastic at the other [59]. The central collagen domain (40–50 kDa) contains multiple Gly-Gly-X repeats, elastin and spider drag-line silk like modules flank this structure. Metal-binding by histidine residues also has roles in intramolecular stabilization of the preColls in the byssus, maturation of adhesive processes and hardening of adhesive assemblies. The byssus preColls have quite different properties to the fibrillar collagens of mammalian tissues which are designed for force transmission in tendons and ligaments and are largely inextensible whereas the preColls are elastic and have force dissipative properties [59,62,63,64]. Adhesive proteins in the plaques of the mussel byssus act in a co-ordinated manner to effect adhesion. Mussel foot proteins act in a concerted fashion to effect adhesion of the byssus. In the initial stages of adhesion Mfp-3 and Mfp-5 act as primers on the adhesive surface and other Mfp’s subsequently bond to this primer layer. Mfp-3 and 5 contain a high L-DOPA content of between 20–30%. L-DOPA undergoes hydrogen bonding to mineral and oxide surfaces and hydrophobic interactions which facilitate its adhesive properties. These interactions are sensitive to oxidation and low pH, lysine located adjacent to L-DOPA on the Mfp backbone may protect L-DOPA and promote adhesion. L-DOPA-lysine repeat modules are common in Mfp proteins. PreCol-NG, a precursor protein and one of three collagenous proteins along with preCol-D and preCol-P with fibroin-like and elastin like flanking domains have structural roles in the mussel byssus thread interactive with lipid vesicles which induces reversible β-structures in these proteins [67]. This shows how lipids can effect cell free self assembly processes in structural fibrous proteins in the mussel byssus thread [68]. Lipids may also regulate the adhesive footprint of the byssus plaque [69] and has also been suggested to displace water from the adhesion surface facilitating Mfp binding to the substrate priming this site for the laying down of other Mfp’s to the attachment site [70]. Information on the byssal adhesive proteins has been used to develop medical bioadhesives of widespread application in healthcare procedures [54,71,72,73,74,75,76,77,78,79,80].

## 3. Barnacle Cement

Barnacle cement proteins are organized into a permanently bonded layer containing nanoscale anchorage fibres [81]. These cement proteins share no homology with any other marine adhesives [82]. A significant portion of acorn barnacle cement is composed of low complexity proteins containing repetitive amino acid blocks homologous to silk motifs with an abundance of Gly/Ala/Ser/Thr repeats such as found in fibroin, silk gum sericin, and spider silk pyriform spidroins. These are assembled into distinct primary structures in a unique class of adhesive nanostructures [83]. It is remarkable that structure function relationships have been established for barnacle cement proteins given the difficulty of obtaining sufficient material for analysis. The adhesive secretions of barnacles are particularly challenging to access, the proteinaceous layer is typically about a micron in thickness and is bonded permanently during the life of the barnacle to the substrate. A recombinant 19 kDa barnacle protein has been produced and characterized [84], silk protein homologies have also been identified in barnacle adhesive proteins [83]. The assembly of barnacle nanofibres have been demonstrated and their roles in barnacle adhesive processes established [85]. Acorn barnacles secrete a phase-separation fluid which primes the surface of the substrate surface prior to deposition of cement proteins to effect adhesion [86]. The adhesive properties of barnacles has aided in the design of adhesives for potential biomedical application [87,88]. Barnacle cement also contains multiple oxidases and proteases which may have roles in the processing and maturation of barnacle adhesive proteins prior during their self-assembly into cement structures [83]. The adhesive proteins of the goose necked barnacle *Pollicipes pollicipes* have also been compared with Acorn barnacle adhesive proteins [89,90] (Table 1).

Barnacle encrustaceans on the hulls of sea-going vessels have been known by seafarers for hundreds of years. Armor-plated acorn barnacles tenaciously are glued to calcareous base plates to rocks, pilings, and foul boat hulls and can even attach to the surface of blue mussels and the skin of whales and other marine species but not to fish which have a protective slimy coating on their skin surfaces. A key aspect of the barnacle adhesion process which differentiates it from other marine adhesives is the initial deposition of a lipid to prepare the prospective adhesive surface [92]. Barnacles secrete a lipid droplet on to the adhesive surface which displaces water and prepares this site for adhesive phosphoproteins secreted by the barnacles which act as a cement [93] (Figure 5g,h). The barnacle adhesive process is therefore a two phase system undertaken by phosphoproteins and lipids contained in separate granules within the cyprid cement glands [94]. Barnacle bio-adhesives therefore differ from adhesive proteins found in mussels and Sandcastle polychaete tubeworms which involve L-DOPA-rich or poly-phosphorylated peptides which become crosslinked at the adhesion site [95,96,97]. Goose necked barnacles utilize a similar adhesive process as Acorn barnacles [98]. The barnacle cement proteins share no homology with any other known marine bio-adhesives. However a recent study has identified that a proportion of the cement proteins contain repeat block gly-ala-ser-thr repeat modules similar to those found in silk fibroin, silk gum sericin and spider silk spidroins and these may form nano-structures important in barnacle adhesion [83]. A recent study using multi-photon and broadband coherent anti-Stokes Raman scattering microscopy [93] delineated the barnacle adhesive structures and the exudates they produced which promoted adhesion.This study showed that barnacle adhesive was a lipid-phosphoprotein bi-phasic system, which acted synergistically to maximize adhesion to a diverse range of aquatic surfaces [93]. These components are contained within two different granules in the cyprid cement glands. Lipids are released first at the adhesion site which displaces water and primes the site for attachment processes with phosphoproteins. A greater understanding of this attachment process opens the possibility of the effective development of anti-fouling strategies which will prevent barnacle attachment to ship hulls in an ecologically friendly and sustainable format.

Fouling of ship hulls reduces speed due to increased drag, results in increased fuel consumption, generation of ozone depleting greenhouse gas emissions and causes corrosive damage to the hull [99,100]. The adhesive properties of barnacles to propellors even under highly agitated arduous binding conditions is stark evidence of the tenacious and extremely rapid binding properties of this mollusk [92,98,101,102,103,104]. Acorn and goose-necked barnacles [89] can both foul ship hulls with the latter imposing a greater impact on the drag characteristics due to its trailing habit whereas acorn barnacles have a more streamlined profile and have less of an impact [99,100,101]. Several countries have introduced legislation mandating that ship hulls must be certified free of fouling organisms in order to gain entry to their territorial waters so development of ecologically permissible effective cleaning or anti-fouling strategies are a high priority for multiple reasons [99,105].

## 4. Sandcastle Worm, Caddis-Fly Fly Larvae Cement

Aquatic arthropods such as Caddis-fly larvae [106,107] and the polychaete marine Sandcastle worm *Phragmatopoma californica* [1,108,109] also employ silks as a cementing material to glue together sand particles or small stones in a precise fashion to produce their protective casings underwater. Tube-building sabellariid polychaetes have major impacts on the geology and ecology of shorelines worldwide. The adhesive produced by the Sandcastle Worm rapidly solidifies to become a weight bearing structure assembling the protective tube-like structures encasing these polychaete worms. The initial fluid glue hardens into a brown, leathery, weight-bearing solid in two phases involving a quick (30 s) initial setting followed by a covalent curing phase over several hours. The final cohesive strength of the Sandcastle adhesive is provided, at least in part, by covalent crosslinking of its adhesive proteins through L-DOPA residues, in a similar manner to the L-DOPA-containing mussel adhesive byssal plaque proteins however phosphoserine residues are a very prominent feature of the Sandcastle worm adhesive and form a strong cementing material similar to that utilized by barnacles in their tenacious binding to marine structures. The Sandcastle worm assembles particles of sand in regular well formed tube formations and will even use small glass beads for this purpose when cultured in-vitro (a) and these illustrate well the very precise manner these are cemented together in tube formations (b).The Sandcastle worm adhesion points on these beads initially have a white appearance (b) but as they cure over time they assume a brownish colouration (c). This adhesive material is autofluorescent and can be visualized by confocal fluorescent microscopy.

*Phragmatopoma californica* Sandcastle cement consists of 3 proteins (Pc1–3) and also contains significant levels of phosphate, calcium, and magnesium minerals [1]. Two of the Sandcastle cement proteins are highly basic proteins (pI 9.7–9.9), Pc-1 (18 kDa) and Pc-2 (21 kDa), [1], and also contain L-DOPA thus they resemble the byssal adhesive proteins of mussels [1]. The Pc-3 protein (30.5 kDa) sequence is dominated by phosphoserine repeat residues with occasional L-DOPA. Phosphoserine and glycine, account for ~60 mol % of all Sandcastle cement protein residues, Pc-3 occurs as seven variants containing 60–90 mol % phosphoserine. Although each worm builds primarily the tube in which it resides, a colony of worms can collectively produce massive boulder-like concretions which protect the sandy shoreline from erosion and has a pivotal role in reef ecology and protection of sandy coastlines.

The Sandcastle worm (*Phragmatopoma californica*), a Pacific Ocean polychaete worm is an inhabitant of the intertidal zone in the shoreline. This worm is encased in a protective casing which it forms by glueing together fragments of shell and sand particles into tube-like structures. In-vitro experiments have also shown it can cement together small glass spheres in perfect architectural form to produce such tube formations. Furthermore adjacent tubes can also be cemented together forming large stabilising formations in sand-banks. Three of the adhesive proteins (Pc1–3) this worm exudes have been sequenced and their adhesive properties characterised [1]. Pc-1 and Pc-2 are small 18–21 kDa proteins containing block copolymer repeat amino acid sequences of 10–14 serine residues punctuated with single tyrosine residues which are modified to phosphoserine and L-DOPA respectively which is an active cross-linking component also found in the mussel byssus attachment zone proteins. Pc-3 consists of a family of at least seven proteins containing 60–90 mol % phosphorylated serine residues which form the tube cement [109,110]. A tough but flexible bio-adhesive inspired by the Sandcastle adhesive proteins based on a biocompatible elastomeric compound formed from poly(glycerol sebacate acrylate), (PGSA) has been developed. These components have been approved by the U.S. Food and Drug Administration for use in humans and can be photo-cured in-situ in 5–30 s. PGSA has excellent biocompatibility, good adhesion and stability with adhesive patches in heart surgery trials in rats and pig tissues and is currently scheduled for human trials. Results with this high performance elastomeric adhesive are eagerly anticipated since it has the potential to replace suturing in surgical operations and may revolutionise such procedures.

The Caddis-fly has a widespread global distribution, Caddis-fly silk peroxinectin (csPxt), a haem-peroxidase, catalyses dityrosine cross-linking within the adhesive peripheral region of the tube layed down around Caddis-fly larvae as protective structures [106,111,112]. Fibroin silk fibres are also components of these tubes with the fibres held together by an adhesive which also cements small stones and bits of vegetation to the external surface of these tubes in a similar manner to the tubes that Sandcastle worms assemble [113]. These tubes are quite stable structures and have even been detected in the Cretaceous fossil record in Western China [114].

## 5. Slug Adhesive

The terrestrial slug *Arion subfuscus* produces a mucus-based defensive secretion that is sticky and tough. Native gel electrophoresis shows *A. subfuscus* glue consists of a network of negatively charged 40–220 kDa proteins and a network of heparan sulfate-like proteoglycans [13]. These two interpenetrating networks, a relatively stiff network and a relatively deformable network, work synergistically to create a tough adhesive material [13]. Upon secretion, selective components in this glue are rapidly oxidized, cross-link formation between the generated aldehydes and primary amines contribute to the cohesive strength of this glue [13]. Formation of metal co-ordination complexes also contribute to the adhesive properties of this gel [115]. Atomic absorption spectrometry shows that the *Arion subfuscus* glue contains substantial quantities of zinc, iron, copper and manganese (2–7 ppm) and these interact with the negatively charged proteins in the gel contributing to the gelation of *A. subfuscus* mucus secretions and its adhesive properties [116]. Two forms of slug mucus are produced one which is non-adhesive and has lubricative properties facilitating slug gliding over surfaces and another form which is strongly adhesive and is secreted when the slug adopts a protective adherent posture when threatened by a predator. Reptilian tongues are also covered by a bi-modal mucus which is adhesive when prey species are being captured but when the tongue returns into the mouth the mucus becomes non adherent and the food item is released [8]. Frog tongues also have specific structural modifications which facilitate this adhesive capture process [10]. Characterisation of the double interpenetrating network slug adhesive [13] has enabled the development of high performance medical adhesives which can be used to seal surgical incisions without the need for sutures, clips or staples [21,117,118]. Slug adhesive is sticky when wet [119]. These adhesives have been shown to efficiently seal demanding tissues such as the beating heart, lungs and mucosal lining tissues [21]. The lack of sutures ensures the surgical site does not leak which can be a source of infection and improves recovery from surgical operations [81,120].

## 6. Gecko Dry Adhesive

Natural adhesives in the feet of different arthropods and vertebrates show strong adhesion, durability and excellent reusability. Hierarchical nano-structures on the surface of foot pads (setae) significantly contribute to adhesion. Development of elastomeric layers with embedded air- or oil-filled microchannels further enhance adhesion (up to 30 fold) of these microstructures through capillary force generated surface stresses. Geckos have evolved one of the most versatile and effective adhesives known in Nature through natural selection forces and have been a source of fascination to scientists for over a century. The astonishing ability of Gecko’s to climb on vertical surfaces is due to the adhesive properties conveyed by fibrous micro-hairs known as setae located on the tips of their toe pads. The adhesive properties setae provide are due to van der Waals interactions generated by the myriads of spatula shaped projections located at the outermost tips of these setae [5]. This micro/nano hierarchical adhesive system is based on structures assembled from keratin fibrils. The angle of the setae end points relative to the main fibril axis is critical not only to the attachment process but also to detachment processes to facilitate non-impeded movement. Machine learning-based computational methods have been developed to optimize the design of these adhesive fibrils [121]. Such novel designs of adhesive fibrils have been produced by two-photon-polymerization-based 3D microprinting and double-molding-replication processes using polydimethylsiloxane as a substrate [121,122,123,124]. These optimal elastomeric fibril designs have been shown to outperform previous fibril assembly methods [122]. Finite-element-analyses shows that fibrillar adhesion is sensitive to the 3D fibril stem shape, tensile deformation, and the precise specifications of the fibril microfabrication process [124]. While investigations on Gecko adhesion have focused on the remarkable dry-adhesive processes conveyed by Gecko foot parts, adhesion can also be achieved under wet high humidity conditions [125,126].

Setae, fibrils located on a gecko’s feet, have inspired the development of synthetic dry microfibrillar adhesives over the last two decades [127]. The unique properties of setae: residue-free, repeatable, tunable, controllable and silent adhesion; self-cleaning; and breathability make them suitable for a wide range of applications hence there is considerable interest in these structures [128]. Bioinspired elastomeric structural adhesives can provide reversible and controllable adhesion on dry/wet and synthetic/biological surfaces for a broad range of commercial applications [25].

The adhesive toe pads of gecko foot structures are modified from the reptilian epidermis. Sticky 10–100 μm filaments or setae are composed of a number of defined proteins termed corneous beta proteins (CBPs) [129]. These are low molecular weight (12–20 kDa) proteins that contain a conserved central region of 34 amino acids. This region forms a beta-conformation that is assembled into long beta-filaments, these undergo aggregation to form corneous beta-bundles which are the basic structural elements of the setae. The CBPs are cysteine-glycine-rich proteins, these are distributed throughout the setae forming structures known as spatulae which have fundamental functional roles to play in Gecko adhesion [129,130]. It has been proposed that interactions between anionic intermediate filament keratins and positively charged CBPs produce resilient flexible corneous assemblies, disulphide bonding may also stabilize these structures. Protein-lipid interactions may also contribute to the pliability of these structures providing compliant properties to the setae improving their adhesion to irregular adhesive surfaces [131]. Nuclear Magnetic Resonance (NMR) spectroscopy also confirms the presence of lipid-keratin structures in the Gecko foot setae [132]. Near-edge X-ray absorption fine structure spectromicroscopy of the structural organization and keratin filament alignments in setae confirms the presence of β-sheet structures and ordered keratin filament alignment with likely roles in the stabilization of setae Gecko foot structures [133].

## 7. Spider Silk

The dragline silk of the Golden Orb-Weaving spider is one of the most studied polymers in science. Spider silk is composed of polymeric modular proteins (Spidroin 1 and 2) of the scleroprotein group which includes mammalian collagens found in fibrillar structures in ligaments and tendon and the keratin protein family found in nails and hair. These are structural proteins which provide mechanical support in various structures. The protein in dragline silk is fibroin (M_r_ 200–300 kDa) composed of the proteins spidroin 1 and spidroin 2. Fibroin consists of approximately 42% Gly and 25% Ala but this varies with spider species and diet. The remaining amino acid components of fibroin include Glutamine, Ser, Leu, Val, Pro, Tyr and Arg. Spidroin 1 and spidroin 2 differ mainly in their Pro and Tyr contents [134]. Silks have a hierarchical structure as a block co-polymer which contains repeat glycine-alanine blocks within an amorphous matrix consisting of helical and beta turn structures (Figure 6). It is the interplay between these hard crystalline β-sheet and strained elastic semi-amorphous helical regions in silk that equips it with extraordinary properties of strength, ductility and elasticity [134,135,136]. The modular structure of silk contains β-spiral, β-sheet and 3_10_ -helices assembled from two peptides, Masp-1 and Masp-2 which have the unusual amino acid repeat sequences GGSGG**Q**GG**Q**GGYGSGG**Q**G**Q**G**Q**GGYGSGAAAAAAAAA (MaSp1).

GXGPGX**Q**GPGX**Q**GGGYGPGAAAAAAAA (MaSp-2). MaSp-1 and MaSp-2 are variably assembled to produce many forms of silk with differing combinations of physical properties (strength, extensibility, elasticity, ductility, resilience). This assembly process occurs in a number of glands in spiders, at least 7 forms of spider web silk have been reported. These include the large ampullate gland which assembles the radial and frame lines in spider webs and the dragline that spiders use for moving around the environs of the web but is separate from the web per-se.The spider also has a piriform gland which produces cement for the peripheral attachment of the web to supportive structures and also reinforces the attachment points between radial and frame lines. The aggregate gland produces sticky ball-like adhesive structures on the fine spiral in the centre of the web which immobilizes insects caught in the web preventing their escape (Figure 6a–d). The frameline and radial lines, the main structural elements of the web and the draglines do not display this sticky feature.

Spidroin contains 4 to 9 polyalanine block repeats interspersed in Gly penta-peptide repeats providing elasticity, β-spirals occur after each these repeats introducing a with a 180° turn in the molecule. Variation in the assembly of these regions provides adjustable properties of strength and elasticity to silk polymers. Spiders synthesisize at least seven forms of silk. Capture silk is the most elastic form containing ~43 Gly repeats and is 200% extensible, dragline silk however only has the Gly repeats repeated nine times and can extend ~30% of its original length. Gly-Gly-Gly-rich repeat segments also give rise to a tight helix which acts as a transitional structure between the polyalanine and spiral regions. During the spinning process these components pass through the spinneret and become aligned and partial crystallisation occurs parallel to the fibre axis through self-assembly [134]. The polyalanine blocks link together via hydrogen bonding forming β-sheet crystalline regions crosslinking silk strands to provide high tensile strength. The abundance of Ala and Gly, the most compact amino acids in silk proteins facilitates dense packing of residues strengthening the molecule. Crystalline regions of silk protein are very hydrophobic aiding in the loss of water and solidification of spider silk explaining why silk is so insoluble-water molecules cannot penetrate the strongly hydrogen bonded β-sheets. The Gly-rich spiral regions of spidroin aggregate to form amorphous elastic regions in the spider silk. Less ordered Ala-rich crystalline regions connect the β-sheets to the amorphous regions providing crystalline regions within an amorphous matrix in a similar manner to how Kevlar is assembled [134]. The motion of a spider down a drag-line silk fibre has been compared with that of a climber descending down an abseil rope. In free space the climber rotates around the abseil rope but the spider does not undergo rotation. This reflects the internal structure of the abseil rope and drag-line silk. Rotation is more pronounced in static cable laid unsheathed abseil rope but less pronounced in a dynamic sheathed climbing rope and does not occur in spider silk. This indicates that the heterogeneous construction of the drag-line silk fibre accommodates those forces which would otherwise have produced a rotational effect on the spider.

### 7.1. Emerging New Spider Web Technologies

Of all of the naturally occurring adhesive and structural materials discussed in this review spider silk adhesives and structural proteins stand out as being some of most advanced materials in terms of performance as mechanically supportive materials and have added properties as biosensory materials which may be of application in emerging new technologies. Generations of scientists, structural and polymer engineers have been fascinated by spider webs and their geometric perfection and the strength of these structures which are achieved with minimalistic design features. This has been instrumental in the design of architectural developments in massive cable supported sporting complexes and stadia globally.

Spider capture silk is an exceptional natural supportive biopolymer that outperforms virtually any man made equivalent synthetic material in terms of its properties of strength, ductility and elasticity. The structure of spider silk is remarkable but is still incompletely characterized despite major advances in the evaluation of structure-function relationships in biomaterials science. X-ray crystallography cannot be used to analyse silk fibres and this has hampered elucidation of its structure. Capture silk in a spider web is the internal spiral in webs containing globules of sticky material that aids in the capture of insect prey. The major protein in the capture threads in spider webs is a flagelliform protein related to silk-worm fibroin. Force microscopy has been used to develop insightful models of the molecular and supramolecular structure of this flagelliform protein based on its characteristic blocks of amino acids and how they behave when they are stretched and pulled in the capture web. Analysis of spider web structure-function relationships [137] has led to the development of biopolymers with predictable structural properties [44,138]. Spider silk fibres display exceptional mechanical features of toughness, elasticity and low density excelling over other natural and man-made fibres and are superior to even Kevlar and steel. These superlative properties stem from long fibre length and specific internal protein structures in the silk fibre which provide mechanical strength as well as elasticity and ductility [134]. Spider silk proteins may have more than 20,000 amino acids with contiguous polypeptide segments accounting for more than 90% of the silk fibre structure [139]. Specific domains in the silk fibre internal structure can be repeated more than a hundred times with each repeat unit having a specific function to play in the composite silk structure. Crystalline rich domains interspersed within amorphous regions provide strength while the amorphous components convey elasticity to the silk fibre [140]. Silk fibres have been used as suturing material in surgical applications for many decades however the aforementioned functional properties make silk an attractive polymer for innovative material development into biomedical products [141,142,143,144,145,146]. One of the problems which has held back research on specific spider silk types is the difficulty of obtaining sufficient quantities for analysis [147]. A major problem in the development of spider silk biomedical products is the harvesting of sufficient material to facilitate such developments. Spider farming is ineffective due to their cannibalistic behaviour. Thus in order to obtain sufficient spider silk proteins (spidroins) methods have been developed to produce recombinant silk protein in expression systems in plants, bacteria, yeasts, insects, silkworms, mammalian cells and animals. For such production systems to be viable large-scale cost-effective and efficient production systems are needed. However due to the incompletely understood complexity of the structure of silk proteins and their assembly into fibre networks and the difficulty in the expression of full-length assembled silk proteins in host organisms this has been a significant challenge. Synthetic silk proteins have been expressed in *E. coli* [148].

Spider silk has also been used as suture material for generations. Fabrication of recombinantly produced spider silk proteins has now opened up modern biomedical applications using spider silk 2D and 3D scaffolds [149]. Spider-web fabrication technology can self-assemble spider proteins as fibers, films, 3D-foams, hydrogels, tubes, and microcapsules as spidroin structures in hydrogels that can be used in deep wound healing, 3D bone repair, oriented fibers for axon growth and nerve tissue regeneration [150]. Spidroin micro- and nanoparticles can also be used as drug delivery and virus-free DNA delivery systems into animal cell nuclei. Such technology can also be used in fabrication of biocompatible, biodegradable soft optic photonic crystal super lenses and fiber optics applications in soft micro-electronic nanogenerator systems and as scaffolds for the growth, proliferation, and differentiation of various types of cells, tissue engineering applications, medical device coating development, soft optics, and micro-electronics [150].

### 7.2. Spider Webs Are Electroconductive and Harvest Atmospheric Moisture

Spider webs are electroconductive through the moisture droplets they harvest from the atmosphere. Electrostatics play a role in insect capture, spider webs are electrostatically attracted to positively charged insects, deforming by a few millimetres towards them with the prey-capture efficiency of the webs enhanced [151]. Because the webs consist of electrically conductive aqueous glue droplets distributed along comparatively insulating silk threads, it has been proposed that a charge separation is induced within the droplets when charged insects fly nearby, creating an attractive force between the web and prey species [152]. Thus the glue droplets on the spider web capture spiral provide additional functional properties to the web [153,154,155,156,157].

### 7.3. Spider Web Silk a Putative Polymer Facilitating Novel Developments in the Design of Biomedical, Micro- and Nano-Electronic Devices and Biosensors

For centuries scientists have attempted to unlock the fantastic capabilities of silk for use in the prospective development of medical devices and biosensors through its ability to impact on microelectronic processes and it’s light-manipulative properties [138,158,159]. Packets of electricity are called electrons; packets of light are photons and packets of sound energy are called phonons [160]. Thus while a semiconductor controls the flow of electrons, spider silk can control the flow of light, sound and heat. Spiders use their webs to receive and send vibratory signals to locate prey and detect defects in their web construction. Silk transmits sounds that the spiders interpret providing a greater spatial understanding of their microenvironment. The sound generated by webs changes frequency depending on how stressed the web is. When the web is compressed the sound frequencies it generates slow down; when webs are strained, these sound frequencies speed up. The crystalline microstructure of spider silk has the potential to unlock a whole new field of nano-engineered materials for use in medicine and engineering. Such phononic silk crystals can also manipulate sound waves. Manipulation of phonons in silk or synthetic polymers designed from these silk structures offer a new research paradigm in photonics and optical science. While semiconductors control the flow of electrons, spider silk has the potential to control the flow of sound and heat, understanding this process will facilitate the development of new “tunable” smart materials for sonic and thermal insulation. Structural engineers can gain insight as to how assembled structures might be non-destructively tested to assess their functional integrity providing predictive capability to avoid failure of such structures. Phononic crystals have the ability to manipulate sound waves and the potential to be developed into triggers, switches or activators to regulate micro and nano-electronic responses [161]. As a biomedical material spider silk has many advantages, it is biodegradable, biocompatible and can be implanted in the body without eliciting an immunological response. Silk can thus be used in biodegradable microelectronics, optical fibres, hologram generation, development of vein and tendon replacements and potentially in developments that will provide super efficient solar cells [162,163].

### 7.4. High Precision Air Filters Based on Spider Proteins

With the development of high precision spinning procedures [164] high performance filters have been prepared from spider silk for the removal of atmospheric micro-particulate matter [165]. These filters outperform existing filtering membranes for the removal of pathogens exhibiting a high efficiency (>99.995% removal of pathogens), low air resistance, high transparency and remarkable bioprotective properties for the removal of biohazardous pathogens [166]. Ultra Net filters are thin (~350 nm) and have electrostatic properties ensuring efficient adsorption of particulate material [167]. These membranes have high mechanical robustness and improved air-flow properties. Carbon nano-nets are electroconductive, titanium dioxide nano-nets have bioprotective properties [168].

Multiparameter integrated sensors for next generation wearable microelectronic devices have been developed using flexible dual-parameter pressure-strain sensors based on 3D tubular graphene sponges and spider web-like stretchable electrodes. Such devices show promise in artificial intelligence and wearable microelectronic applications [165]. Mechanically stable 3D porous high conductivity graphene multiwalled carbon nanotube-silicone rubber composite high sensitivity wearable high accuracy medical monitoring intelligent robot perception systems have been developed to improve healthcare monitoring [169].

## 8. Caterpillar Silk

Silk-worm silk is a composite biomaterial composed of two main proteins, fibroin and sericin. Fibroin provides strong and elastic properties in spun silk structures, while sericin proteins are glues which seal the silk filaments into a single fibre and provide fibre stickiness needed for assembly of the cocoon. A further silk protein was discovered in the silk of the wax moth, *Galleria mellonella*, in 1998 called seroin which has anti-microbial properties [170]. Two seroins were subsequently identified as small glycoproteins of 22.5 and 23 kDa with identical N-termini, 64 seroins have subsequently been identified from 32 Lepidopteran species. Microsequencing of their C-termini has uncovered conserved regions and identified 5 seroin sub-families in the *Lepidoptera* composed of three kinds of structural domains. Seroin proteins contain 20 amino acid structural peptide modules in a mature protein up to 250 amino acid residues long.

### The Silk Produced by Silk-Worm Larvae Bombyx Mori

Fibroin is an insoluble insect silk protein produced in larval developmental stages [171]. Silk fibroin is a β-keratin like protein related to the keratin found in hair, skin, nails and connective tissues. The silk produced by the silk-worm larvae *Bombyx mori*, the domestic silk-moth, differs from that of spider silk spidroins in that it is produced as a central double fibroin strand, a second protein, sericin surrounds the fibroin in three layers [172] (Figure 7). Fibroin is composed of light, heavy, and a glycoprotein P25 polypeptide chains [173]. The heavy and light chains are disulphide linked and P25 associates with these by noncovalent interactions providing stabilization to the complex [174]. The primary structure of fibroin is rich in Gly-Ser-Gly-Ala Gly-Ala repeat modules which form into anti-parallel β-sheet structures.The high Gly and to a lesser extent Ala content facilitates tight packing of the silk polypeptide chains contributing to its stability and strength with its material properties being of interest in tissue engineering [173,174,175,176] and biomedicine [177,178,179,180,181]. Fibroin occurs in three structural forms in silk I, II, and III. Silk I contains the unprocessed natural form of fibroin [182] secreted by the *Bombyx mori* silk glands, Silk II is a spinned silk [182] and Silk III is a form that occurs at liquid-air interfaces [183]. *Bombyx mori* silk is composed of 60–80% fibroin, 15–35% sericin and 1–5% non-sericin components. For two decades, the fibroin component of silk has been the major focus in the search for medical biomaterials and biomedical applications and the sericin component was ignored or abandoned as a waste byproduct during the processing of traditional silk fabrics or silk biomaterials. However, sericin is also a highly useful biological material [184,185,186]. A further family of non-sericin component proteins, the seroins also have interesting anti-microbial and anti-viral properties [170,187,188,189,190].

## 9. Medical Adhesives

Appropriate acknowledgement should be made of the seminal work of Prof J Herbert Waite in the early catechol adhesive literature and in studies they have inspired [191,192,193,194]. This work has made a significant contribution to the development and commercialization of dental bioadhesives and the widespread application of bioadhesives in general in many biomedical applications [87,195,196].

The number of surgical procedures conducted worldwide is considerable and continues to grow every year. An estimated number of over 300 million surgeries were conducted in 2012 [197,198], and 310 million in 2020 [199]. Surgical closure techniques include sutures, staples or clips however these may result in secondary tissue damage, microbial infection, fluid or air leakage, and disfigurement. The use of sutures in delicate tissues such as vascular anastomosis, nerve repair, or ocular surgeries is technically demanding and have associated risk factors.The development of a robust medical adhesive to obviate suturing procedures represents a significant technical advance, speeding up these procedures, reduces the chances of infection and improves the healing of wounds and post-operative recovery of the patient. Medical adhesives are spread over the entire area of the surgical wound site avoiding the concentration of point stresses at suture points which can tear in delicate tissues forming a flexible closure which seals more effectively and is less likely to fail. Three forms of tissue glues can be distinguished (i) haemostats, (ii) sealants, and (iii) adhesives. These have differing properties, a haemostat promotes blood clotting and reduces blood loss but does not function in the absence of blood. A sealant produces a barrier layer preventing leakage of fluid or gas from the surgical site and an adhesive binds two tissue surfaces firmly together.

Medical adhesives are promising alternatives to sutures and staples in a large variety of surgical and clinical applications. Bioadhesives which are currently in use include FloSeal^®^, CoSeal^®^, BioGlue^®^, Evicel^®^, Tisseel^®^, Progel™ PALS, and TissuGlu^®^. These can be used in combination with albumin, glutaraldehyde, chitosan, cyanoacrylate, fibrin and thrombin, gelatin, polyethylene glycol (PEG) and urethanes with each formulation customised for a specific application [200]. Marine or terrestrial animal inspired adhesives we have discussed in this review have greater adhesive properties, are non-toxic, biodegradable and do not elicit an immunological response thus show exceptional promise in biomedical applications. Inspired by the granule-packaged glue delivery system of Sandcastle worms, a nanoparticulate formulation of a viscous hydrophobic light-activated adhesive based on poly(glycerol sebacate)-acrylate has been developed. Negatively charged alginate stabilizes the nanoparticulate surface reducing its viscosity and injectability through small-bore needles. Aqueous nanoparticulate glues of 30 *w*/*v* % dispersions remain localized at injection sites. Use of a positively charged polymer (e.g., protamine), rapidly initiates assembly of nanoparticulate glues into an adhesive polymer that can augment or replace sutures and staples during minimally invasive surgical procedures [201]. Poly(glycerol-co-sebacate acrylate) elastomer based adhesives modify the prospective attachment surface mimicking the nanotopography of gecko feet and is aided by their biocompatibility, biodegradation, strong adhesive properties and their compliance and conformability to tissue surfaces [202]. These adhesives can also deliver drugs or growth factors to promote healing. This gecko-inspired tissue adhesive prepared from a biocompatible and biodegradable elastomer was optimized by varying dimensions of the nanoscale pillars, including the ratio of tip diameter to pitch and the ratio of tip diameter to base diameter [202]. Coating these nanomolded elastomer pillars with a thin layer of oxidized dextran significantly increased the interfacial adhesion strength on porcine intestine tissue in vitro and in the rat abdominal subfascial environment in-vivo. This gecko-inspired medical adhesive may have applications in sealing wounds and replacement or augmentation of sutures or staples thus may be of widespread application. An elastomeric bioadhesive with impressive adhesive properties has also been developed from the mucous exudate of the slug *Arion subfuscus* [21].This tough elastic adhesive adheres extremely efficiently to an extremely difficult target tissue, the beating heart, and has the potential to revolutionise surgical approaches on cardiac and mucosal tissues.

While the primary objective of a bio-adhesive is to act as a wound sealant or as a haemostatic agent to stem blood loss, the development of multi-layered tissue adhesive sealants [203] offers the opportunity of incorporation of growth factors, and slow release antibiotics or anti-inflammatory drugs into such wound closure devices with their breakdown in-situ releasing their entrapped components accelerating wound healing or elevating the efficacy of tissue regeneration [204,205]. Advanced performance surgical tapes inspired by natural bio-adhesives have also been developed with BMP binding peptides incorporated, this formulation improves growth factor binding at wound repair sites and improves healing in ACL reconstruction [43]. Various therapeutic drug formulations could also be incorporated offering long-term pain relief during wound healing. Poly(amino acid) bioadhesives show potential application in wound closure in abdominal and heart surgery/tissue repair applications [21,206] and have superior performance to conventional fibrin based adhesive formats currently used in orthopaedic and other surgical applications [207,208]. Use of L-DOPA bioadhesive formulations as a therapeutic cell delivery vehicle to apply stem cells to focal cartilage lesions in OA using arthroscopic procedures offers exciting possibilities. The superior adhesive properties of L-DOPA adhesives to wet cartilage surfaces suggests a likely positive outcome for cell delivery in this potential application. Mussel foot protein-5 has already been employed to improve attachment of cultured cells [209,210]. The CS and pullulan based adhesives which have been used for therapeutic cell delivery to cartilage defects in repair strategies [211,212] have inferior adhesive properties compared to such L-DOPA based bio-adhesives [207,213]. Recombinant mussel foot proteins are available commercially [214,215,216] and are compatible with cell culture conditions [210,214] and these offer exciting possibilities in cartilage repair and regeneration strategies which deserve to be evaluated experimentally.

## 10. Future Research with Bioadhesives

Various therapeutic drug formulation incorporations in bioadhesives offer long-term pain relief during wound healing of difficult painful wounds such as chronic venous diabetic ulcers which have a notoriously poor healing capability and are extremely painful [217]. Bio-adhesive scaffolds even show potential as mucosal and intranasal vaccine delivery systems [218,219,220]. As already discussed poly(amino acid) bioadhesives show potential application in wound closure in abdominal and heart surgery/tissue repair applications [21,120,206] and have superior performance to conventional fibrin based adhesive formulations currently used in orthopaedic and other surgical applications [207,208]. Use of L-DOPA bioadhesives as a therapeutic cell delivery vehicle to apply stem cells to focal cartilage lesions in OA using arthroscopic procedures offers exciting possibilities.The superior adhesive properties of L-DOPA adhesives to wet cartilage surfaces suggests a likely positive outcome for cell delivery in this potential application. Mussel foot protein-5 has already been employed to improve attachment of cultured cells [210,214]. Chondroitin sulphate and pullulan based adhesives have been used for therapeutic cell delivery to cartilage defects in repair strategies [211,212] however these have inferior adhesive properties compared to L-DOPA based bio-adhesives [207,213]. Recombinant mussel foot proteins are available commercially [214,215,216] and are compatible with cell culture [210,214] and these offer exciting possibilities in cartilage repair and regenerative strategies which deserve to be evaluated experimentally. Double-sided adhesive tape inspired by spiderwebs and gecko dry adhesive proteins could revolutionize surgical interventions by simplifying closure of surgical incisions in difficult tissues and also providing faster recovery rates [221]. Silk fibers have been used to form miniscule bio-friendly dome lenses utilizing the liquid-collecting capability of the web capture spiral. Solidified dielectric dome lenses with different dimensions prepared using ultraviolet curing may be useful in bio-photonic applications [222]. Silk fabrics have been used in the production of laminates in anti-stab body-armor and have superior stopping power for knife penetration than Kevlar which is primarily designed for protection from bullet penetration [223]. Dragon-silk has also been developed and used in the production of body armour and has superior bullet stopping power than Kevlar [224]. A mussel-inspired double-crosslink wet tissue adhesive has been developed using a L-DOPAmine-conjugated gelatin macromer, Fe^3+^ initial crosslinker, and genipen maturational crosslinker. This catechol-Fe^3+^ mediated crosslinking provides a controllable instant adhesive with significantly higher wet tissue adhesive properties than commercial fibrin glue on wet porcine skin and cartilage [206]. This adhesive is elastic, biodegradable, and biocompatible and is suitable for internal tissue adhesion, sealing, and hemostasis applications [225,226,227]. A catechol-conjugated chitosan bioadhesive with similar properties has also been developed for biomedical applications [225,227]. These adhesives are advocated for internal biomedical applications [226,227] and as smart bioadhesives for wound healing and closure [80].

## Figures and Tables

**Figure 1 molecules-27-08982-f001:**
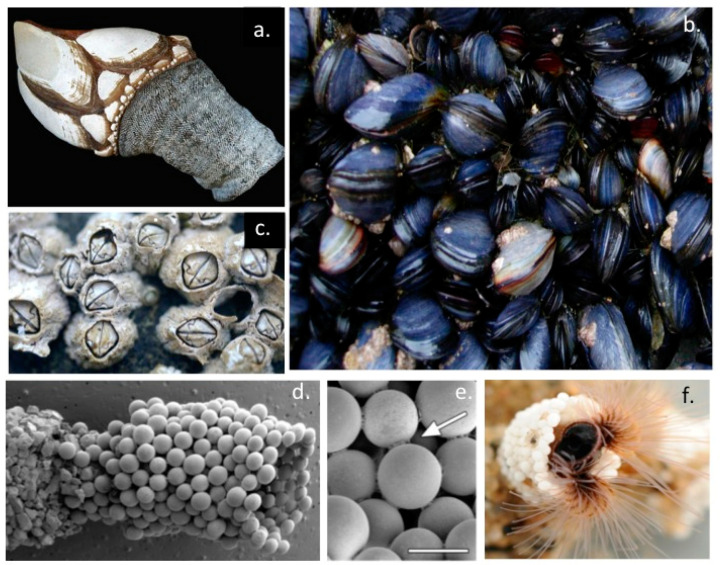
Some marine organisms that secrete adhesive proteins that they use to adhere to rocks and other marine surfaces. Gooseneck barnacle *Pollicipes pollicipes* (**a**), blue mussel *Mytilus edulis* (**b**), Acorn barnacle *Chthamalus stellatus* (**c**), and the sabellariid polychaete Sandcastle worm *Phragmatopoma californica*, showing the detailed assembly of its tube-like structures using micro glass beads supplied in the laboratory rather than the sand particles used in nature (**d**), intricate point adhesions used to cement glass beads togethor are labelled with small white arrow, scale bar 100 μm (**e**) and a sandworm inhabiting a tube (**f**). Images **a**–**d** from Wikipedia Commons under Open Access Creative Commons Attribution-Share Alike 4.0 International licence, © Hans Hillewaert 2006. Images **d**–**f** reprinted with permission from Ref [1] under Open Access.

**Figure 2 molecules-27-08982-f002:**
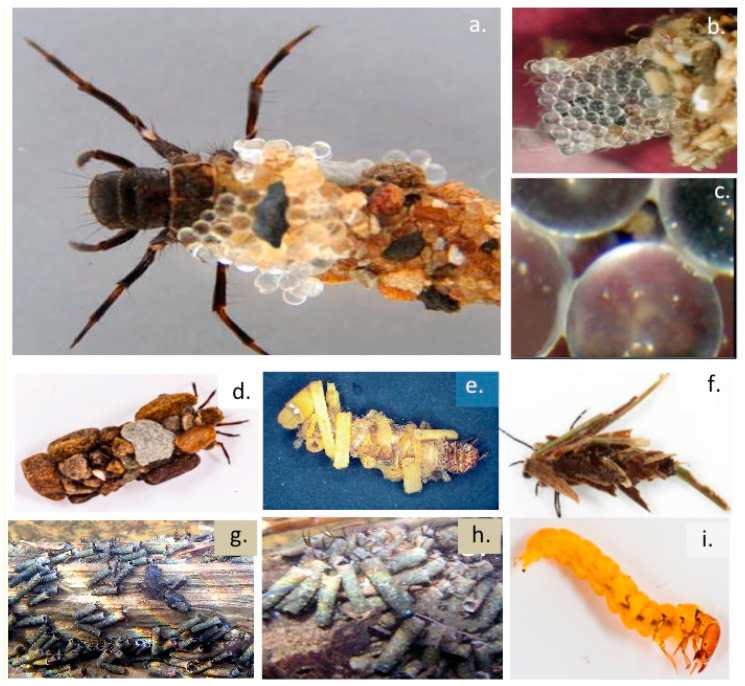
A Caddis-fly larva in the underwater encasing structure it lays down (**a**) of similar construction to the tube-like structure assembled by the Sandcastle marine worm and some examples of this tube assembled from microglass beads in the laboratory (**b**,**c**). Small white arrows depict attachment points between beads (**c**) and small stones and pieces of vegetation in nature (**d**–**f**). Arrangements of Caddis-fly fly tubes attached to the underside of stones are also shown (**g**,**h**). A Caddis-fly larva removed from a tube assembly is shown (**i**). Images reproduced from [2] with permission and also supplied by Dr Joyce Gross, UC Berkeley, Natural History Museum.

**Figure 3 molecules-27-08982-f003:**
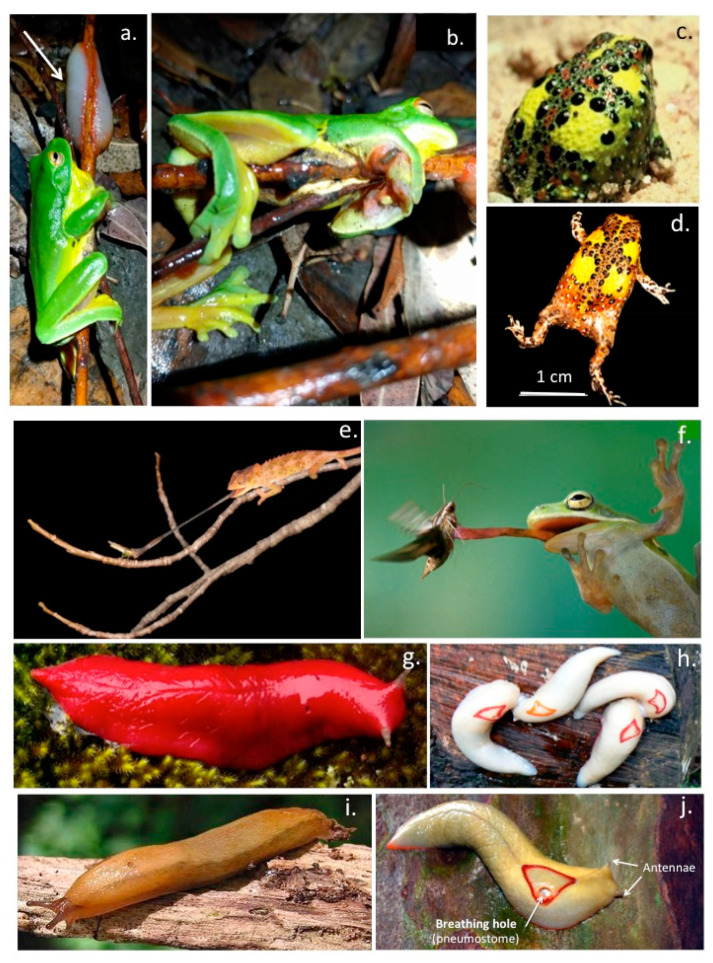
Examples of the red eyed green tree frog *Ranoidea chloris* found along the eastern seaboard of Australia, immobilised by an adhesive exudate (white arrow) (**a**,**b**) secreted by the red triangle slug Triboniophorus graeffei. The crucifix toad *Notaden benetti*, also known as the holy cross frog (**c**,**d**) which exudes a strong adhesive through its dorsal skin that results in prolonged sexual union when mating. Examples of the elongated tongue of a chameleon and red eyed green tree frog capturing a prey insect (**e**,**f**). The tongue is covered in a mucinous adhesive secretion which aids in capture of prey items. The distinctive large scarlet slug of Mount Kaputar, Northern, New South Wales, Australia (**g**) and the related smaller elaborate red triangle slug (**h**). The former inhabits the summit regions of Mount Kaputar (elevation 1500 m) while the latter is found in the tropical rain-forests at lower altitudes. The dusky slug *Arion subfuscus* whose sticky exudate has been developed into a useful surgical bioadhesive (**i**). Details of the breathing hole (pneumostone) and antennae of the red-triangle slug (**j**). Images **a**,**b** courtesy of Dr John Gould, School of Enironmental and Life Sciences, University of Newcastle, Australia, and images **c**,**d** by Prof Mike Tyler, Dept of Zoology, The University of Adelaide. Image e supplied by The Australian Museum. Image f was an Alamy stock photograph. Images **g**–**j** supplied courtesy of The Australian Geographic Magazine and Australian Museum. Image **i** by Tom Meijer/Erik Veldhuis was obtained from Wikipedia Commons under Open Access CC BY-SA 3.0, http://creativecommons.org/licenses/by-sa/3.0/ (accessed on 2 February 2022).

**Figure 4 molecules-27-08982-f004:**
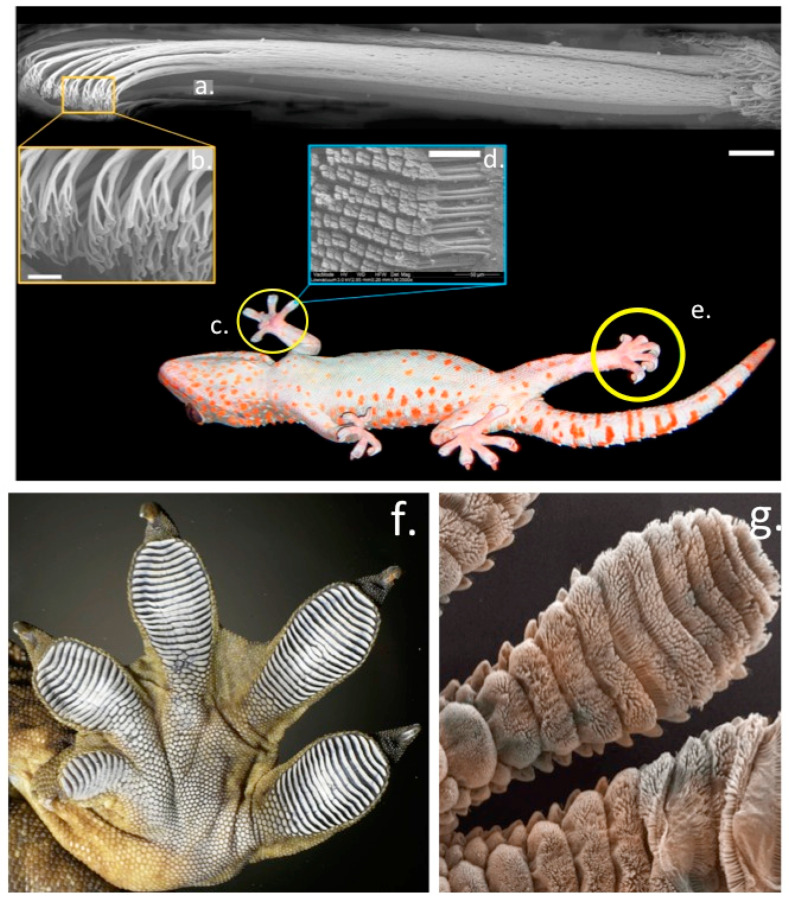
The structural organisation of the Gecko foot parts which facilitate remarkable dry adhesion properties on vertical surfaces. Bundles of fine hair-like setae aligned in the toes of the foot parts visualised by electron microscopy, notice the terminal specialisation in these structures which is critical to the adhesive process (**a**–**e**). Higher power images of the foot parts showing the aligned rows of setae (**f**,**g**). Images reproduced from [30] with permission.

**Figure 5 molecules-27-08982-f005:**
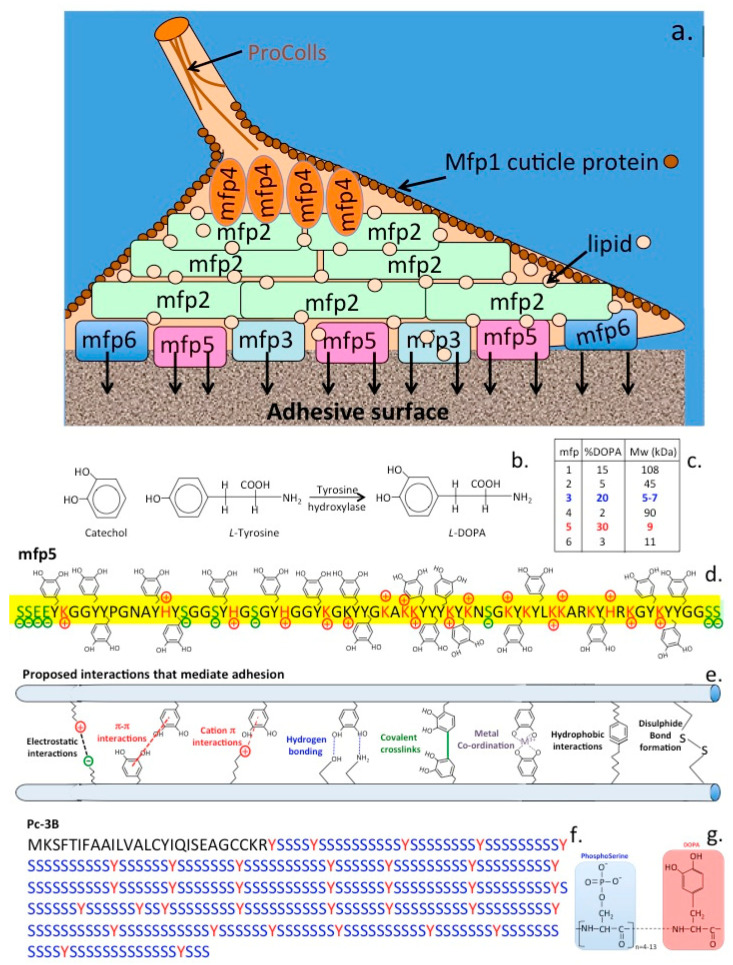
Schematic depiction of the mussel byssus adhesive proteins (Mfp’s) showing the hierarchical organisation of Mfp-2-6 in the attachment zone and the cuticle forming Mfp-1 (**a**). Pre-col proteins in the byssal thread have visco-elastic properties of importance to the dynamic dissipation of forces through this structure. The catechol moiety of L-DOPA has important roles in byssal adhesion and is produced by the the action of tyrosine hydroxylase on L-tyrosine in specific compartments of the mussel during fixation (**b**). The Mfp’s have variable contents of L-DOPA (**c**). Mfp-5 with a high L-DOPA content (**d**) has particularly important contributions to make to byssal adhesion by a number of mechanisms based on the catechol functionality of L-DOPA (**e**). Barnacles also utilise L-DOPA to facilitate adhesion to substrates however is distinguished from the mussel attachment process through interactions mediated by phosphoserine which is a more predominant barnacle amino acid and has roles in the formation of permanent concrete-like attachment to sub-strata (**f**,**g**).

**Figure 6 molecules-27-08982-f006:**
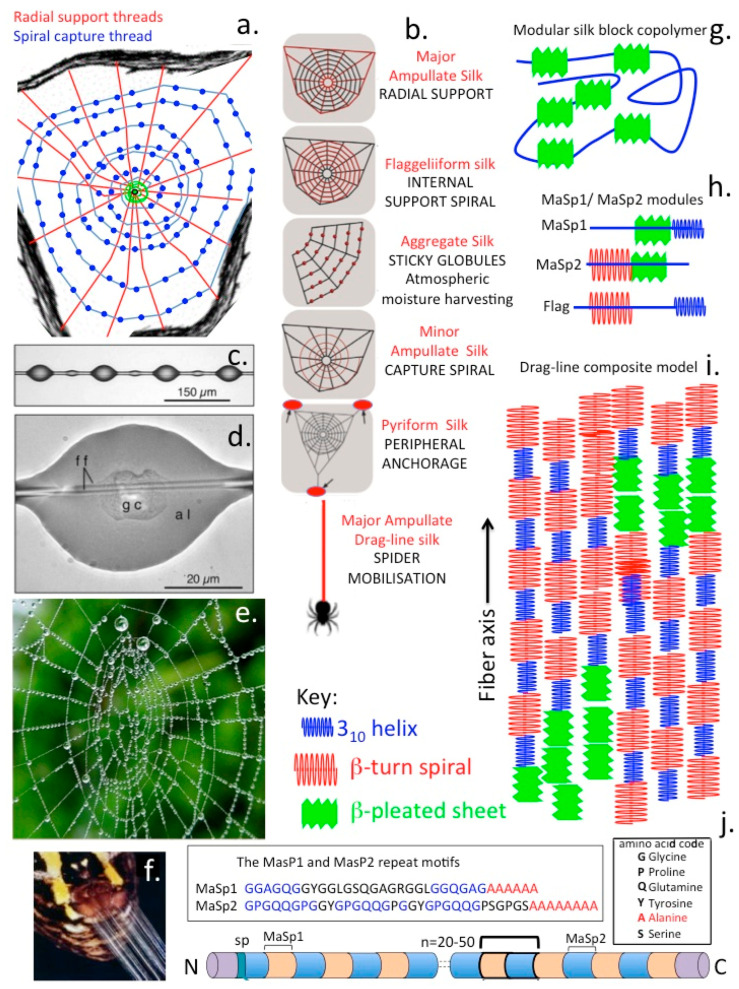
Composite schematics figure depicting the complexity of spider silk assembly into web structures with inserted images depicting the sticky globules of the capture spiral and how moisture is harvested from the atmosphere as water droplets distributed in these globules in the web and secretion of silk fibres from a spider spinerette gland. The radial support fibres distributed like spokes in a wheel and the radial capture spiral and its globular deposits are major features of web assembly (**a**). Different silk fibre types assembled into specific structural features identified in the web, including ampullate silk fibres in the radial support fibres, flaggeliform silk fibres in the internal support spiral, aggregate silk of the sticky globules of the support spiral, minor ampullate silk of the capture spiral, pyriform silk deposits as peripheral anchorage regions of the web and major ampullate drag-line silk that the spider uses to move around the web complex (**b**). Sticky globules assembled in the capture spiral showing internal central glycoprotein (gc) depositions (**c**,**d**) responsible for retrieval of atmospheric moisture (**e**) and capture of prey species. High power image showing silk fibre ejection from a spinerette which are spun into fibres of variable composition (**f**). Major structural features of functional modules in the composite silk fibre showing its modular block copolymer construction (**g**) and the β-pleated sheet modules (**h**) which assemble as micro-crystalline regions, and 3_10_ helices, and β-turn spirals in the MaSP1/MaSP2 spidroin components of the composite silk fibre (**i**). Crystalline β-pleated sheet regions are considered to convey strength to the fibre and the 3_10_ helices, and β-turn spirals convey elasticity and ductility to the silk fibre. Modular amino acid repeat sequences in silk fibre spidroin protein (MaSP1/MaSP2) components of the composite silk fibre showing their block amino acid repeat polyalanine modules and GGAGQGG modular repeats of Spidroin1 (MaSP1) and GPGQQGP modules of Spidroin2 (MaSP2) (**j**).

**Figure 7 molecules-27-08982-f007:**
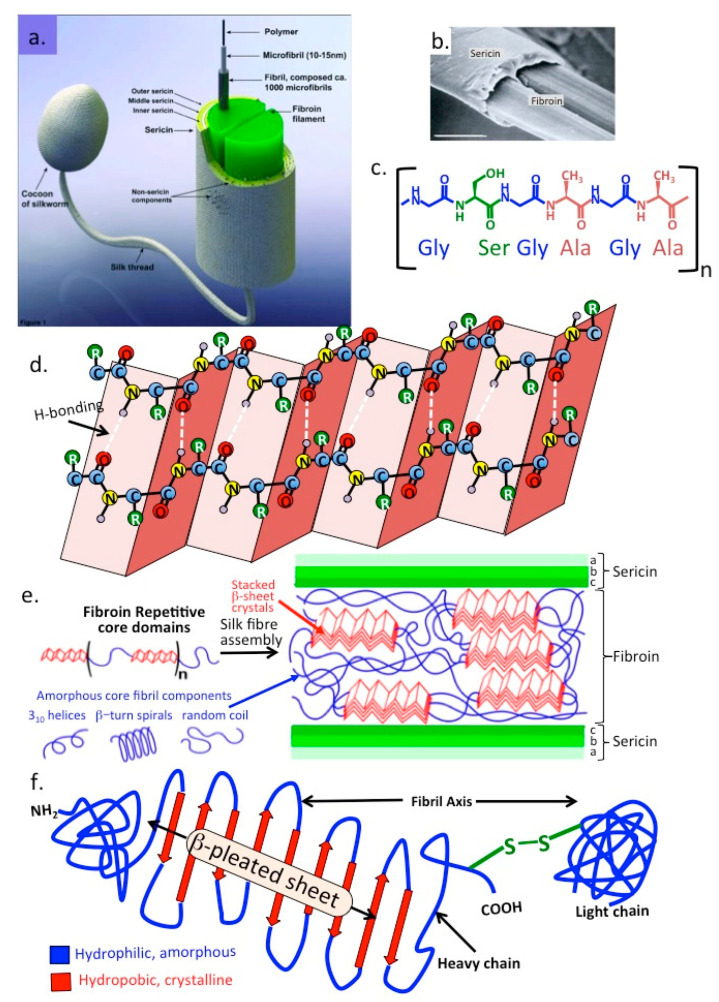
Schematic depiction of the silk fibres produced by silk-worm larvae showing the central two strands of fibroin surrounded by a multi layer of sericin (**a**,**b**) and the repeat amino acid module found in sericin (**c**) and β-pleated sheet structures in fibroin that stack together forming micro-crystalline regions which provide mechanical strength to the silk fibre (**d**). These micro-crystalline regions are interspersed within amorphous regions in fibroin containing 3_10_ helices, β-turn spiral random coils that provide elasticity (**e**). Primary structure of fibroin showing its three polypeptide N-terminal, central β-pleated sheet heavy chain and disulphide bonded C-terminal chain containing hydrophilic and hydrophobic regions with important functional roles in fibroin’s structural organization (**f**). Image (**b**) depicts an electron micrograph of fibroin strands surrounded by sericin.

**Table 1 molecules-27-08982-t001:** Fully sequenced adhesive proteins listed in the National Centre for Biotechnology *.

Common Name	Species	Protein	NCBI Entry	Ref.
Barnacle	*Megabalanus rosa*	Mrcp-19k	BAE94409	[91]
		Mrcp-20k	BAB18762	[92]
		Mrcp-52k	BAL22342	[93]
		Mrcp-100k	BAB12269	[94]
	*Balanus albicostatus*	Balcp-19k	AB242295	[91]
		Balcp-20k	AB329666	[95]
	*Balanus improvisus*	Bicp-19k	AB242296	[91]
Spider	*Nephila clavipes*	ASG1	EU780014	[96]
		ASG2	EU780015	[96]
		PySp2	HM020705	[97]
	*Latrodectus hesperus*	AgSF1	JX262195	[98]
		PySp1	FJ973621	[99]
Mussel	*Dreissena polymorpha*	Dpfp1	AAF75279	[100]
		Dpfp2	AM229730	[101]
	*Mytilus californianus*	Mfp-3S	DQ165556	[102]
		Mcfp-5	DQ444853	[103]
		Mcfp-6	DQ351537	[104]
	*Mytilus edulis*	Mefp-1	AY845258	[105]
		Mefp-3	AF286136	[105]
		Mefp-5	AAL35297	[106]
	*Mytilus galloprovincialis*	Mgfp1	D63778	[107]
		Mgfp5	AY521220	[108]
	*Perna viridis*	Pvfp-1	AAY46226	[109]
		Pvfp-2	AGZ84282	[97]
		Pvfp-3	AGZ84285	[97]
		Pvfp-5	AGZ84279	[97]
		Pvfp-6	AGZ84283	[97]
Slug	*Lehmannia valentiana*	Sm40	ABR68007	[110]
		Sm85	ABR68008	[110]
Tubeworm	*Phragmatopoma californica*	Pc-1	AAY29115	[111]
		Pc-2	AAY29116	[111]
		Pc-3A	AY960618	[111]
		Pc-3B	AY960621	[112]
		Pc-4	GH160602	[112]
		Pc-5	GH160603	[112]
	*Sabellaria alveolata*	Sa-1	CCD57439	[113]
		Sa-2	CCD57460	[113]
		Sa-3A	CCD57480	[113]
		Sa-3B	CCD57502	[113]

* Information modified from [91].

## Data Availability

Not applicable.

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
