# Peer review of "High Performance Marine and Terrestrial Bioadhesives and the Biomedical Applications They Have Inspired"

_molecules, 2022, doi:10.3390/molecules27248982_

Round 1
Reviewer 1 Report
This is a remarkable review about high Performance Marine and Terrestrial Bioadhesives. The explanations of how they work are very clear.
Only minor revisions needed to be done.

Author Response
Response to Reviewer 1 comments
Only minor typos noted- these have been corrected
Reviewer 2 Report
The review summarizes very thorougly the latest data on the topic. It is very well written, gives a complete picture of the subject, and outlines the future basic studies and clinical applications. The review will be of great interest for a wide range of scientists, both working in this topic and those who want to get a first idea about the subject.
However, I would suggest some minor corrections before the acceptance of the MS.
1. First of all, proofreading is required to correct some typos etc:
sandcastle (e.g. l 53), Sandcastle (l 73) or Sand Castle (l. 470)?-
Caddis fly (l 77), caddis fly (68) or Caddis-fly (l 473)
a typo: proteinaceous (l 120)
In figures legends, the font size differs within a legend (Figure 2 - ll 139-153, Figure 5 - ll 284-290)
Mfp's (l 230) or mfps (l 261) for plural?
Dopa, L-Dopa (both versions in Chapter 2), dopa (l 267) or L-DOPA (l 443)
et cetera
2. ll 413-417 seem to be irrelevant to the subject of the review
4.
Author Response
Responses to Reviewer 2
The items indicated have all been corrected and are highlighted in the Mkd up revised version of the manuscript
According to my information proteinaceous is the correct spelling
Reviewer 3 Report
Dear Author, this reviewer is an adhesive specialist but very far from your field of adhesives. I must congratulate you for such a wonderful review, very nice, very interesting and very well presented. There are just some very minor additions that I would like you to make in your reference list. These are:
1. Waite, JH. Evidence for a repeating 3,4-Dihydroxyphenylalanine and hydroxyproline-containing decapeptide in the adhesive proptein of the Mussel, Mytilusedulis, L., The Journal of Biological Chemistry, 258(5), 2911-2915 (1983).
2. "Waite JH, Andersen NH, Jewhurst S, Sun CJ. Mussel adhesion: Finding the tricks worth mimicking. J Adhesion. 2005:297–317."
3.H. J Waite, “Mussel adhesion - essential footwork.”, J Exp Biol, vol. 220, no. Pt 4, pp. 517-530, 2017.
4. Wei Wei, Luigi Petrone, YerPeng Tan, Hao Cai, Jacob N. Israelachvili, Ali Miserez, and J. Herbert Waite, Adv Funct Mater. 2016 26(20), 3496–3507.
There are many other papers of Prof. Waite, from really long time ago as I did know of his work when he asked me in the 1980's about catechol units effect in bioadhesives. Considering his fundamental work on the subject, and the fact that his work has led to commercialization of dental bioadhesives I think you should mention a bit more of him a you talk in your paper of catechol moieties linked to protein sequenes (your figure 5). Do not try to guess who I am, you will not find my name anywhere linked to the one of Dr Waite.
For the rest, wonderful review. Well done!!
Author Response
Response to Reviewer 3
Thank you for your kind comments, such comments are important to authors. I agree with your comments on J Herbert Waite and have added the following segment that is highlighted in the Mkd up version of the revised manuscript.
Appropriate acknowledgement should be made of the seminal work of Prof J Herbert Waite in the early catechol adhesive literature and in studies they have inspired [1-4]. This work has made a significant contribution to the development and commercialization of dental bioadhesives and the widespread application of bioadhesives in general in many biomedical applications [5-7].
- Waite JH. Evidence for a repeating 3,4-dihydroxyphenylalanine- and hydroxyproline-containing decapeptide in the adhesive protein of the mussel, Mytilus edulis L. J Biol Chem. 1983 10;258(5):2911-5.
- Waite JH, Andersen NH, Jewhurst S, Sun CJ. Mussel adhesion: Finding the tricks worth mimicking. J Adhesion. 2005:297–317.
- Waite JH. Mussel adhesion - essential footwork. J Exp Biol. 2017 ;220(Pt 4):517-530.
- Wei W, Petrone L, Tan Y, Cai H, Israelachvili JN, Miserez A, Waite JH. An Underwater Surface-Drying Peptide Inspired by a Mussel Adhesive Protein. Adv Funct Mater. 2016 ;26(20):3496-3507.
- Gan K, Liang C, Bi X, Wu J, Ye Z, Wu W, Hu B. Adhesive Materials Inspired by Barnacle Underwater Adhesion: Biological Principles and Biomimetic Designs. Front Bioeng Biotechnol. 2022 Apr 25;10:870445.
- Sani ES, Lara RP, Aldawood Z, Bassir SH, Nguyen D, Kantarci A, Intini G, Annabi N. An Antimicrobial Dental Light Curable Bioadhesive Hydrogel for Treatment of Peri-Implant Diseases. Matter. 2019 Oct 2;1(4):926-944.
- Duan W, Bian X, Bu Y. Applications of Bioadhesives: A Mini Review. Front Bioeng Biotechnol. 2021 Sep 3;9:716035.